# Vulnerability of the Ancient Peat Plateaus in Western Siberia

**DOI:** 10.3390/plants10122813

**Published:** 2021-12-19

**Authors:** Alexander Pastukhov, Tatiana Marchenko-Vagapova, Sergey Loiko, Dmitry Kaverin

**Affiliations:** 1Institute of Biology Komi Science Centre Ural Branch Russian Academy of Sciences, Kommunisticheskaya 28, 167982 Syktyvkar, Russia; dkav@mail.ru; 2Institute of Geology Komi Science Centre Ural Branch Russian Academy of Sciences, Pervomaiskaya 54, 167982 Syktyvkar, Russia; timarchenko@mail.ru; 3BIO-GEO-CLIM Laboratory, National Research Tomsk State University, Lenina 36, 634050 Tomsk, Russia; s.loyko@yandex.ru

**Keywords:** cryolithozone, degradation, bogs, arctic peatlands, permafrost, soil–geocryological complex, palynological spectra, macrofossil composition, georadar survey

## Abstract

Based on the data of the plant macrofossil and palynological composition of the peat deposits, the evolution and current state of polygonal peatlands were analyzed at the southern limit of continuous permafrost in the Pur-Taz interfluve. Paleoreconstruction shows that peat accumulation began in the Early Holocene, about 9814 cal. year BP, in the Late Pre-Boreal (PB-2), at a rate of 1 to 1.5 mm year^−1^. Intensive peat accumulation continued in the Boreal and early Atlantic. The geocryological complex of polygonal peatlands has remained a stable bog system despite the predicted warming and increasing humidity. However, a rather rapid upper permafrost degradation and irreversible changes in the bog systems of polygonal peatlands occur with anthropogenic disturbances, in particular, a change in the natural hydrological regime under construction of linear objects.

## 1. Introduction

Peatlands, occupying only 3% of the land surface, contain about 15–30% of global soil organic carbon reserves [1], thereby playing a significant role in the regulation of general planetary processes, such as biogeochemical and biogeophysical cycles, greenhouse gases, and activity and species diversity of vegetation and soil biota. In Western Siberia, peatlands cover 592,440 km^2^, exceeding 50% of all regional area in the taiga zone, and have the peat thickness up to 8–10 m [2]. The Western Siberian Plain stands out among other boreal plains by phenomenal bogging, which has both global and regional significance. The polygonal peatlands and peat plateaus are the most extensive bog types in the study area [3]. However, in the tundra, polygonal peatlands with a thick peat deposit (from 2 to 5 m and more) are not uncommon [3,4,5].

Current climate warming and wetting in Siberia led to a lake area increase by 0.89% detected for the 1999–2013 period and an increase by 4.15% for the 1999–2018 period. In Eastern Siberia, a lake area expansion trend was detected in high yedoma fraction areas, indicating ongoing Yedoma degradation by lake thermokarst [6]. In western Siberia, climate warming, increased precipitation, and permafrost thaw are also accompanied by an increase in the frequency of full or partial drainage of thermokarst lakes. After lake drainage, highly productive plant communities on nutrient-rich sediments may develop, thus increasing the influencing greening trends of Arctic tundra [7]. Thermokarst features, such as thaw ponds of thermokarst lakes, are hotspots for methane emissions in warming lowland tundra [8].

In contrast to the northeastern European tundra, where both the climatic optima of the Atlantic and Subboreal of the Holocene, and, at present, permafrost has been thawing due to warming [9], in the tundra zone of Western Siberia, even the maximum warming occurred within the range of negative mean annual soil temperatures and, thus, did not lead to the permafrost retreat [5,10]. However, very-high-resolution (VHR) images showed various types of disturbances over permafrost areas’ extensive networks of hydrocarbon exploration and infrastructure occurring in the Yamal Peninsula in the last several decades, stimulating the initiation of new thermokarst features [11,12]. The significant warming and seasonal variations of the hydrologic cycle, in particular, increased snow water equivalent acting in favor of deepening of the active layer; thus, an increasing intensification of the processes of thawing of underground ice wedges and destruction of soil–geocryological complexes of polygonal peatlands [13]. Peatlands are the most stable from the standpoint of the temperature state of permafrost [14], but the high amount of ice wedges in soil–geocryological complexes makes them very vulnerable to anthropogenic impacts [15]. Our earlier studies in the European Northeast with the use of high-frequency, ground-penetrating radar showed that the construction and operation of a road embankment with a hard cement-concrete coating crossing peat plateaus in the southern permafrost was limited, leading to the permafrost table retreating down to 8 m, and the warming effect of the road construction affected the field 50 m wide [16].

Nowadays, there is increasing attention to the study of peat genesis and properties, since peat supports and affects bog ecosystems, which have a unique communities’ structure and a high biodiversity level [4,10,17,18]. However, publications on the features of the permafrost peatlands’ evolution are extremely insufficient [7,19,20,21,22] and spore-pollen spectra studies are limited [4,23,24]. Significant reserves of soil organic carbon are conserved as a peat. Therefore, the peat pool plays an important role in the biogeochemical carbon cycle and climate change processes [25]. Peat monoliths are archives of information on paleoenvironmental conditions [26].

The aim of the study was to characterize the evolution and current state of ancient peat plateaus (polygonal peatlands), at the southern limit of the continuous permafrost in Western Siberia, as well as to analyze their vulnerability to anthropogenic impact.

## 2. Results

### 2.1. Landscape Settings and Plant Communities

The studied peatland is a fissured, convex-polygonal, shrub-lichen on polygons, and dwarf-sphagnum bog in interpolygonal depressions, bounded on both sides by thermokarst lakes (Figure 1). The peatland has polygons 10–40 m in diameter. The polygons are in different development stages, and occupy from 80 to 90% and more of the total bog area. As a natural one, the polygonal peatland is almost flat from the surface. The excess of polygons in depressions, representing wide cracks, is no more than 0.5 m. Polydominant shrub-lichen and shrub-ledum-lichen phytocenoses are widespread at the polygons. The portion of peat circles (bare peat patches) is insignificant, less than 1% of the peatland. The peat circles with a diameter of 0.5–2 m are developed mainly at the polygon edges. The polygons’ surface is ombrotrophic, having typical but commonly sparse and depressed bog vegetation. *Ledum palustre*, *Betula nana,* and *Rubus chamaemorus* are dominant shrubs, less often with *Oxycoccus microcarpus* and *Empetrum hermaphroditum*, *and* occasionally are found *Vaccinium vitis-idaea* and *V. uliginosum*. The projective vegetation cover of each of these species varies from 5% to 25%. The ground layer is dominated by lichens, over 70%. The most abundant and constant species are *Cladonia rangiferina*, *C. arbuscula*, *C. stellaris*, *Flavocetraria nivalis*, and *F. islandica* (total coverage up to 40%). The portion of other lichens is much lower. The abundance of mosses does not exceed 30%. *Polytrichum strictum*, *Pleurozium schreberi,* and *Dicranum elongatum* mostly grow in these communities. Sphagnum mosses (*Sphagnum fuscum* and *S. compactum*) are rare in occurrence. In the interpolygonal depressions, shrub-grass-sphagnum phytocenoses dominate. The grass-dwarf shrub layer is represented mainly by *Betula nana* and *Eriophorum media*, as well as by *Rubus chamaemorus* and *Oxycoccus microcarpus*, and, in the ground layer, *Sphagnum fuscum* and *S. compactum*.

### 2.2. The Current State of the Permafrost of the Polygonal Bogs

Permafrost peatlands are typical bog geosystems in permafrost environments in Canada [27], Scandinavia [28], European Russia [19], and Siberia [4,5]. Permafrost initiation involves peatland surface upheaval that results in drying of the peat surface, which is often prone to abrasion and erosion processes [29]. Scandinavian palsa mires [30] on Eastern European peat plateaus [3] occur at the marginal zone of permafrost distribution. Therefore, they may react rapidly to small changes in climate conditions such as warming and increasing precipitation. However, even with this, the small active layer thickness is caused by seasonal variations in the heat conductivity of the surface peat, which protects the soil–geocryological complexes of permafrost peatlands from thawing [31]. In contrast to palsas, the Western Siberian Arctic peatlands occur in much more severe environmental conditions. Even a significant (by 10–15 °С) climate warming during the Holocene Atlantic optimum occurred on a vast area within the limits of negative temperatures and did not lead to the permafrost degradation. The climatic conditions of this epoch did not allow thawing of Arctic permafrost peatlands, reinforced with ice wedges. Ice wedges did not thaw from the surface even in the southernmost part of the Yamal Peninsula (67° N) [5].

As a result of the construction and operation of a bulk road with a cement-concrete coating, the peatland hydrological regime changed rapidly. The processes of thermal corrosion and thermokarst resulted from ice wedges thawing in the interpolygonal depressions. In this case, the polygon edges collapsed, and residual peat mounds developed. The height of such polygonal “mounds” relative to the bottom of the interpolygonal depressions might exceed 3.5 m. Polygonal peatlands were transformed into peat plateau complexes, which clearly proves the erosion hypothesis of their origin, formulated by N.I. Pyavchenko [32,33]. Polygons located close to the road have a slightly different vegetation cover and structure of communities. Shrubs are taller and thicker. *Betula nana* and *Salix phylicifolia* grow on the slopes of the destroyed polygons. In the vegetation cover, the role of *Eriophorum media* increased significantly. The ground layer is composed mainly by mosses (*Sanionia uncinata*, *Pleurozium schreberi*, *Polytrichum strictum*). Deep cracks and depressions are waterlogged or filled with peat, occasionally covered by sedge communities (*Carex limosa* or *C. paupercula*, less often *Carex chordorrhiza* and *C. rotundata*).

### 2.3. Soils and Structure of the Peat Strata of the Polygonal Peatland

According to the International Soil Reference Base [34], the soils of the polygons are classified as Ombric Sapric Cryis Histosols (Hyperorganic), the characteristic diagnostic features of which are the presence of well-decomposed organogenic material (peat), predominantly of atmospheric nutrition with a thickness of more than 2 m and the occurrence of a permafrost table within 1 m. Hemic Muusic Histosols have moderately decomposed peat located directly above the polygonal-veined ice, filling the interpolygonal cracks. Ombric Sapric Cryic Histosols (Hyperorganic Turbic) are formed on peat circles (bare patches). In contrast to the polygon, the soils of the peat circles are cryoturbated in the upper peat horizon with the processes of exfoliation and structuring of peat.

Below are descriptions of peat soils in polygons and interpolygonal cracks in the study area, as well as an analysis of the plant macrofossils and spore-pollen composition of the soil horizons.

The TZ profile is located at the polygon edge with a recently thawed ice wedge (Figure 1; Appendix A). The height of the polygons relative to the bottom of the thawed crack exceeds 3.5 m. The crack, starting from a depth of 2.3 m, is filled with water and peat that has fallen from the edges of the polygon. The vegetation cover is shrub-lichen and consists of *Betula nana*, *Ledum palustre*, *Rubus chamaemorus*, lichens, and hypnum mosses. Peat circles occupy up to 10% of the site. The active layer depth on 15 August 2017 was 35 cm.

The studied polygonal peatland is composed mainly of *Ledum*-hypnum, *Ledum*-sphagnum, and grass-sphagnum peat with an admixture of dwarf shrubs (Figure 1). At the initial stage, on the site of the polygon, there was a grass- and sedge-moss eutrophic community dominated by *Equisetum* sp., *Carex limosa*, *C. rotundata*, and *Menyanthes trifoliate*. Periodically, the moisture conditions changed and hypnum mosses replaced sphagnum for a long time. Only the upper 20 cm of peat strata are mainly composed of mesotrophic dwarf shrubs, which are also common for present-day vegetation (*Salix* sp., *Betula nana* and species of Ericaceae).

The stratigraphic description of the soil–geocryological complex TZ indicates six stages of its genesis (Figure 2):VI—oligotrophic: *Ericales (Ledum palustre + Vaccinium uliginosum) − Dicranum* sp. + *Polytrichum* sp.V—eutrophic: *Menyanthes trifoliata − Sphagnum warnstorfii + S. centrale*,IV—eutrophic: *Equisetum + Menyanthes trifoliata − Sphagnum warnstorfi*i,III—eutrophic: *Equisetum + Menyanthes trifoliata − Sphagnum warnstorfii + Calliergon* sp.,II—eutrophic: *Equisetum + Menyanthes trifoliata − Sphagnum warnstorfii*,I—eutrophic: *Equisetum + Menyanthes trifoliata + Carex limosa − Sphagnum warnstorfii + Meesia*

The TZf profile is located at the interpolygonal cracks near the TZ profile (Figure 1; Appendix A). The vegetation cover is shrub-lichen and consists of *Betula nana*, *Rubus chamaemorus*, *Vaccinium oxycoccos*, *Eriophorum vaginatum,* and sphagnum mosses. The active layer thickness on 15 August 2017 was 35 cm.

The studied interpolygonal depression is composed mainly of sphagnum and herbaceous peats (Figure 3). At the initial stage, on the site of the depression, there was an herbaceous and grass-sedge eutrophic community dominated by *Equisetum* sp., *Eriophorum* sp., *Menyanthes trifoliata*, *Scheuchzeria palustris*, *Carex limosa*, and *C. chordorrhiza*. Then, obviously, sharply drier and colder conditions set in, and herbaceous phytocenoses first were replaced by oligotrophic *Sphagnum* (*Sphagnum lenense*), later by cushy-sphagnum communities (*Sphagnum lenense*, *S. balticum*, *Eriophorum* sp.) and, finally, by mesotrophic shrub-sphagnum plant communities (*Sphagnum balticum*, *Ericales*).

The stratigraphic description of the soil–geocryological complex TZf indicates four stages of its genesis (Figure 3):IV—mesotrophic: *Ericales − Sphagnum balticum*,III—transitional: *Eriophorum* sp. – *Polytrichum* sp. + *Dicranum* sp.,II—oligotrophic: *Eriophorum* sp. − *Sphagnum lenense*,I—eutrophic: *Equisetum + Menyanthes trifoliate + Scheuchzeria palustris + Carex limosa*.

### 2.4. Spore-Pollen Zones of Peat Strata

Palynological analysis of 48 samples collected from the TZ profile indicated the presence of pollen and spores of a good preservation degree. There were also mineral, plant, and post-pyrogenic carbonaceous remains, with sponge spicules. Seven spore-pollen zones were identified totally (Figure 4).

*Spore-pollen zone I* (11 samples at 2.7–2.1 m; 9814–9350 cal. yr. BP; Appendix B: Figure A1). The total composition was dominated by pollen of trees and shrubs (60–83%). Herbaceous plants were found to be more than 35% and spores were 2–7%. Among trees, small-leaved species constituted the main share, e.g., species of genus *Betula* L.: *Betula* sect. *Albae* (over 36%), *Betula* sect. *Fruticosae* (over 22%), *Betula nana* (up to 6.6%), and *Betula* sp. (9%). The group of conifers was represented by *Picea* sp. (up to 19%) and *Pinus sylvestris* (up to 4.8%). Pollen of herbaceous plants was dominated by Cyperaceae (up to 14.7%). Vacciniaceae-Ericaceae, Poaceae, *Artemisia* sp., and Chenopodiaceae were also found. Grass was represented by Apiaceae (up to 7%), Polygonaceae, Rosaceae, Ranunculaceae, Rubiaceae, and Saxifragaceae. The hydrophilous Menyanthaceae was noted (up to 2%). Spore plants were represented by *Sphagnum* sp. (up to 4%), ferns of Polypodiaceae, and *Lycopodium* sp.

*Spore-pollen zone II* (three samples at 2.1–1.87 m; 9350–9250 cal. yr. BP). The total composition was dominated by pollen of trees and shrubs (about 78%). The share of pollen of herbaceous plants was more than 23%. Spores were almost 4%. Among trees and shrubs, species of the genus *Betula* L. predominated, but their proportion was decreasing: *Betula* sect. *Albae* (up to 19.6%), *Betula* sect. *Fruticosae* (13%), *Betula nana* (up to 4.4%), and *Betula* sp. (up to 7.5%). In the group of conifers, the share of *Picea* sp. was up to 33.6%. *Pinus sylvestris*, *P. sibirica*, *Alnus* sp., and *Salix* sp. were singles. The pollen composition of herbaceous plants was still dominated by Cyperaceae (almost 12%). Poaceae, Vacciniaceae-Ericaceae, *Artemisia* sp., and Chenopodiaceae were also found. Among grass, the species of Ranunculaceae, Polygonaceae, Rosaceae, Caryophyllaceae, Apiaceae, and the hydrophilous Menyanthaceae (2.6%) were noted. Few spore plants were found, e.g., Polypodiaceae, *Sphagnum* sp., and *Lycopodium* sp.

*Spore-pollen zone III* (10 samples at 1.87–1.3 m; 9250–8898 cal. yr. BP). The total composition of the spectra was still dominated by the pollen of trees and shrubs (up to 80%), while herbaceous pollen is about 32%. The amount of spores was up to 6.7%. *Betula* sect was dominant, e.g., *Albae* (up to 30%), *Fruticosae* (almost 26%), and *B. nana* (6%). In the group of conifers, the abundance of *Picea* sp. decreased significantly (up to 15%), while *Pinus sylvestris* and *P. sibirica* were single. The pollen composition of herbaceous plants was still dominated by Cyperaceae (up to 8%), but its amount was decreasing. Poaceae, Vacciniaceae-Ericaceae, *Artemisia* sp., and Chenopodiaceae were found. Grass was represented by Asteraceae (up to 6.6%), Ranunculaceae, Polygonaceae, Rosaceae, Apiaceae, and Menyanthaceae. Spore plants were dominated by Sphagnum sp. (3.4%). Polypodiaceae and *Lycopodium* sp. are rare.

*Spore-pollen zone IV* (five samples at 1.3–1.02 m; 8898–8000 cal. yr. BP). In the total composition of the spectra, the pollen of trees and shrubs still prevailed (up to 82%), while herbaceous pollen was about 25% and spores were up to 5%. Despite the predominance of *Betula* L., the share of conifers significantly increased, e.g., *Picea* sp. (up to 31%) and *Pinus sylvestris* (up to 13%). The share of pollen of *Betula* sect. *Fruticosae* was up to 23%, while *B.* sect. *Albae* and *B. nana* decreased to almost 13% and to 6% respectively. *Alnus* sp. and *Salix* sp. were found. Among herbaceous plants, Vacciniaceae-Ericaceae was predominant (10.6%) but Cyperaceae decreased to 8%. Poaceae, *Artemisia* sp., and Chenopodiaceae were noted. Grass was represented by Apiaceae, Rosaceae, Ranunculaceae, Polygonaceae, Onagraceae, and Saxifragaceae. Among spore plants, Polypodiaceae, *Sphagnum* sp., and *Lycopodium* sp. were noted.

*Spore-pollen zone V* (15 samples at 1.02–0.2 m; 8000–4800 cal. yr. BP). The total composition of the spectra was dominated by pollen of trees and shrubs (about 80%). Herbaceous pollen was up to 35.6%, while spore plants were almost 6%. Among trees, the main species’ composition was preserved. The dominant groups were *Betula* sect. *Albae* (25.5%), *B.* sect. *Fruticosae* (25.8%), and *Picea* sp. (almost 16%). Pinus sylvestris, Betula nana (up to 8%), *Alnus* sp., and *Salix* sp. were found in small quantities. The composition of herbaceous pollen became more diverse. Cyperaceae reached 14.5%. Vacciniaceae-Ericaceae, Poaceae, xerophytes: *Artemisia* sp., and Chenopodiaceae were found. Among mesophilic grass, Apiaceae (12.6%), Rosaceae (6%), Ranunculaceae (almost 5%), Polygonaceae, Primulaceae, Polemoniaceae, Brassicaceae, and Menyanthaceae were noted. Spore plants were rare, e.g., *Sphagnum* sp. (up to 5.5%), *Lycopodium complanatum*, *L. annotinum*, and *L. pungens*.

*Spore-pollen zone VI* (two samples at 0.2–0.1 m; 4800–2500 cal. yr. BP). The total composition was dominated by pollen of trees and shrubs (almost 82%), while herbaceous plants ranged from 8% to 30% and the share of spores varied from single values to 50%. Among trees, small-leaved species were still the most widespread, e.g., species of the genus *Betula* L.: *B.* sect. *Albae* (up to 28%), *B.* sect. *Fruticosae* (almost 25%), and *B. nana* (increased to 16%). *Picea* sp. varied from 5% to almost 22%. *Pinus sylvestris* was above 5%, while *P. sibirica* and *P. Abies* sp. were singles. *Alnus* sp. and *Salix* sp. were noted. The composition of herbaceous plants, in general, throughout the entire zone was preserved. Sedges still dominated (up to 24%), and their participation was noticeably increasing, e.g., Vacciniaceae-Ericaceae (about 6%). Poaceae, *Artemisia* sp., and Chenopodiaceae were noted. Grasses were represented by Ranunculaceae, Rosaceae, Polygonaceae, and Apiace. Among spore plants, *Sphagnum* sp. dominated, reaching almost 50% in one sample of all encountered forms. Spores of Polypodiaceae and *Lycopodium* sp. were noted.

*Spore-pollen zone VII* (two samples at 0.1–0 m; 2500 cal. yr. BP—Present). The total composition of the spectra was dominated by pollen of trees and shrubs (76%). Herbaceous pollen was almost 17%. Among trees, *Betula* sect. *Albae* still prevailed (up to 25%), while *B.* sect. *Fruticosae* was almost 19% and *B. nana* decreased to 2–8%. *Picea* sp. was more than 21%. *Pinus sylvestris*, *P. sibirica*, *Alnus* sp., and *Salix* sp. were found. Herbaceous plants contained pollen from Cyperaceae (up to 12.6%), Vacciniaceae-Ericaceae, Poaceae, and *Artemisia* sp. The participation of mesophilic grass decreased. Among spore plants, sphagnum mosses dominated, reaching almost 50% in one sample of all encountered forms. Spores of Polypodiaceae, *Lycopodium clavatum*, *L. complanatum*, *L. annotinum*, and *L. pungens* were noted.

Palynological analysis of 19 samples collected from the interpolygonal depression (TZf section) indicated the presence of pollen and spores of a good degree of preservation. There were also mineral, plant, and post-pyrogenic carbonaceous remains, with sponge spicules. Two spore-pollen zones were identified (Figure 5).

*Spore-pollen zone IIa* (two samples at 1.0–0.9 m). The total composition of the spectra was dominated by the pollen of trees and shrubs (about 70%), but the share of herbaceous plants was insignificant (up to 9%). Spores varied from 12% to 40%. Among trees, the genus *Betula* L. predominated, e.g., *B.* sect. *Albae* (17.5 to 25%), *B.* sect. *Fruticosae* (10–15%), and *B. nana* (5–7%). The group of conifers was represented by *Picea* sp. (up to 16.7%), *Pinus sylvestris* (up to 3%), and *P. sibirica*. *Alnus* sp., *Salix* sp., and *Alnus alnobetula* Subsp. *fruticosa* were singles. Among the encountered herbaceous plants, only Cyperaceae was noticeable (up to 6–9%). *Sphagnum* sp. dominated among spore plants (up to 40%). Polypodiaceae, *Lycopodium clavatum*, and *L. annotinum* were rare.

*Spore-pollen zone I* (4 samples at 0.9–0.7 m). The total composition of the spectra was mainly dominated by the pollen of trees and shrubs (above 75%). Spore plants dominated in two samples (50.6% and almost 76%). Herbs made up 2–12%. In the group trees, species of the genus *Betula* L. generally prevailed, e.g., *B.* sect. *Albae* (almost 24%), *B.* sect. *Fruticosae* (up to 15%), and *B. nana* (up to 7%). The pollen of herbaceous plants was rare, e.g., the predominant Cyperaceae was only 7.4%, while Vacciniaceae-Ericaceae, Poaceae, *Artemisia* sp., and Chenopodiaceae were singles. Apiaceae, Rosaceae, Polygonaceae, Ranunculaceae, and Menyanthaceae were found among mesophilic grass. Spore plants were dominated by *Sphagnum* sp. (up to 76%). Polypodiaceae, *Lycopodium clavatum*, *L. complanatum*, *L. annotinum*, and *L. pungens* were rare.

*Spore-pollen zone II* (13 samples at 0.7–0 m). In the total composition of the spectra, the share of pollen of trees and shrubs dominated and reached almost 71%. Herbaceous plants varied from 16% to 38%, spores ranged from 6–30%. Small-leaved species of the genus *Betula* L. still predominated, e.g., *B.* sect. *Albae* (9–13%), *B.* sect. *Fruticosae* (up to 12%), and *B. nana* (12.5%). *Picea* sp. strongly decreased from 4% to single values, while the share of *Pinus sylvestris* increased to 26%. *Alnaster* sp. (up to 3%), *Abies* sp., *Alnus* sp., and *Salix* sp. were found. The composition of herbaceous plants was preserved in general throughout the entire zone. Vacciniaceae-Ericaceae (about 27%) noticeably increased, while Cyperaceae (6%) and *Rubus chamaemorus* (up to 7% in some samples) were found in fairly large numbers. Among grass, Ranunculaceae, Rosaceae, Polygonaceae, and Asteraceae were noted. Among spores, *Sphagnum* sp. predominance was significantly reduced to 6–29%. Polypodiaceae and *Lycopodium* sp. were also found.

### 2.5. Ground-Penetrating Radar Survey

On the radarogram of the GPR profile 1 in the PK 0–2 segment, the in-phase axis was clearly traced, reflecting the changes of the permafrost table surface (Figure 6). If, under the conditions of an undisturbed polygonal peatland, the permafrost table depth varies in the range of 0.4–0.5 m (markers 2–3), while approaching the road embankment, the permafrost table retreats to a depth of 1 m and below (markers 0–1). Thus, the active layer thickness increases under the road embankment warming effect.

In interpolygonal depressions, the permafrost table is 10–20 cm lower relative to the main surface under the native conditions of undisturbed peatland (Figure 7, PK 3). The depressions are distinguished by a stronger watering of the soil profile, which contributes to increased thermal conductivity and, accordingly, a greater depth of seasonal thaw.

## 3. Discussion

Studies on the Western Siberia region show that the active process of peat accumulation started in the Preboreal stage. Bogs were formed as a result of the end of the Sartan glaciation and the beginning of climate warming, but their development proceeded in local sites in well-pronounced microdepressions [10]. As with the oldest and largest peat deposits in the Eastern European forest-tundra and tundra [21,36], the studied peatland TZ began to develop as a result of the paludification of mixed birch and coniferous woodlands earlier than 9814 cal. yr. BP at a rate of 1 mm per year during the Preboreal (PB) (10,300–9000 cal. yr. BP) (Figure 3). No other ancient dates were obtained for a thick peatland (peat deposit exceeds 5 m) developed on the high sea terraces of the Gydan Peninsula, 9940 cal. yr. BP at a rate of 0.7 mm/year [4]. The severe continental climatic conditions of the Late Dryas changed to a milder and humid climate [37], and the bog system development began with the overgrowing and peat formation of lakes of various genesis, which were mostly thermokarst with submerged taliks. During the second half of the Preboreal (PB, zone I, 9580–9350 cal. yr. BP), birch-spruce woodlands predominated in the study area. Groups of shrub and dwarf birches played a great role, while pine was an admixture. Bog–tundra plant communities are widespread.

At the beginning of the Boreal (BO-1, zones II, IV), the climate became warmer and more humid. There was a gradual increase in the role of spruce and the participation of *Betula* sect. *Albae* had more than halved. The presence of dwarf and shrub birches was greatly reduced. In open sites, bog–herbaceous communities were widespread, with a significant increase of sphagnum mosses. The rate of peat accumulation reached 1.5 mm/yr., up to the end of the Boreal climatic optimum. However, after 8898 cal. yr. BP (BO-2, zone III), there was a reduction in forest vegetation and an increase in groups of dwarf and shrub birches, alder, and willow. At the same time, the rate of peat accumulation was reduced by more than two times, to 0.7 mm/year.

During the entire Atlantic (AT, zone V) (8000–4800 cal. yr. BP), spruce–birch forests predominated in the study area. The forest area periodically increased and then decreased depending on climate fluctuations. In open sites, fen–herbaceous plant communities were developed. If, at the beginning of the Atlantic period, the peat accumulation rate reached 1 mm/yr., then, from 7749 cal. yr. BP, it was about 0.26 mm/yr., and from 6801 to 5657 cal. yr. BP it was only 0.13 mm/year, after which it almost stopped. In the Subboreal (SB, zone VI) (4800–2500 cal. yr. BP), there was a decrease in taiga and an increase in areas occupied by dwarf thickets of *Betula* sect. *Fruticosae* and *B. nana*. Herbaceous associations of sedges and mesophilic grasses were developed in fens.

In the Subatlantic (SA, zone VII) (2500 cal. yr. BP—present), spruce–birch woodlands are widespread, which alternates with extensive fens. The data of the plant macrofossil composition indicate the transition of the eutrophic stage of the bog transformation into the oligotrophic stage, accompanied by a change in the species’ composition of paleocommunities (the replacement of fen grass–moss peats with dwarf shrub–hypnum peats). The possible mechanisms for that could not be explained by climatic changes but was a result of restorative succession after vegetation cover disturbances under local factors (fires, since carbonaceous residues are recorded, and/or changes in hydrological conditions due to permafrost heaving) in the surrounding area.

Thus, paleoreconstruction shows that intensive peat accumulation began as early as the Preboreal, when the temperature and moisture content were even lower than contemporary ones, and the slowing to almost complete cessation of peat accumulation occurred already in the middle of the Atlantic period, when the maximum climatic optimum still went on. For such arctic peatlands, a fairly clear temporal limitation of the period of their most active vertical growth was noted, from 9000 to 6000 years ago, with a rate reaching 1.5–4.4 mm/year. These environmental conditions (due to a unique combination of temperature and moisture) had been no longer repeated either in the second half of the Atlantic period or later [5]. The irregularity of the peat accumulation rates is explained both by active soil heave [38] and following peat cryoturbations, as well as by the predominance of texture-forming, congelation, and wedge-shaped ice in the peat soil, when the mass fraction of biogenic material (peat) is commonly less than 10% of the total mass [5]. In the interpolygonal depression (site TZf, Figure 5), an inversion of two radiocarbon dates was observed: The median age of peat at a depth of 70–75 cm is 7885 cal. yr. BP, and at a depth of 90–95 cm—520 cal. yr. BP. Obviously, this inversion is explained by the result of the peat turbation due to the formation and growth of an underground ice wedge. Thus, in the studied site, a space of 60–65 cm was filled with a horizontal ice vein, expanded to 20 cm on the left wall, and turned into a large block of ice wedges, over 3 m.

In contrast to the tundra and forest–tundra of the European Northeast, where peat accumulated in non-permafrost bogs covered with woody plants, sedges, and mosses and permafrost appeared only at the beginning of the Subboreal (4600–4300 cal. yr. BP), after which, during warming, the degradation of permafrost repeatedly occurred, in the West Siberian tundra, even the maximum warming of the Atlantic optimum climate occurred within negative temperatures and did not lead to a significant degradation of the permafrost table. Model calculations show that by 2050 yr. in Western Siberia, due to global warming, the air temperature may rise by 0.9–1.5 °C and the humidity will increase by 12–39% [39]. Similar climate changes occurred during the Atlantic (about 8000 cal. yr. BP), and in such a climate the peatlands absorbed significant amounts of atmospheric carbon, with peat accumulation rates reaching 0.5 mm/yr. [40]. Therefore, under natural conditions, it can be assumed that the soil–geocryological complexes of studied and other Arctic peatlands will remain a stable carbon sink. It is believed that the West Siberian peatlands in the 21st century will not only remain a carbon sink, but will also increase its absorption [41]; however, the methane emission will obviously also increase [42]. Taking into account the higher Global Warming Potential of CH_4_ compared to CO_2_ (28–34 times) [43], increased methane emissions can significantly affect the overall impact of thawing permafrost, the degradation of arctic peatlands, and climate change.

Under the present-day climate warming, the vegetation cover of polygonal peatlands and peat plateaus protects the permafrost from thaw. When the surface of polygons and peat plateaus dries up, moss ground layer is replaced by lichens and peat circles are formed. However, dry peat provides thermal insulation, preventing further permafrost thawing. Such peatlands might collapse due to wind abrasion and thermal erosion, but thermokarst processes are uncommon there. The thawing of peat plateaus and polygonal peatlands occurs in case they are destroyed or under hindered surface runoff, when lake and fen formation occur. The main indicator of bog stability is the average long-term ground water level, and for permafrost peatlands, the thickness of the active layer is also equal to the averaged long-term depth of seasonal thawing of peat deposits [44]. Since the water levels in the bogs are close to the soil surface, any anthropogenic linear objects crossing them would change the bog runoff. Therefore, the construction of the road favored the flooding of the studied peatland, which led to a change in the hydrothermal regime and vegetation cover and an increase in the temperature of the upper permafrost as well as soil-geocryological complex degradation. Thus, technogenic impacts exceeded the potential stability of the studied peatland as a bog system, and, therefore, irreversible destruction processes may occur.

The analysis by ground-penetrating radar studies showed that the zone of the warming effect of the road as a result of changes in the hydrological regime of the investigated peatland already exceeds 50 m. The most significant retreat of the permafrost table (up to 2–3 m) was observed directly at the bottom of the slope of the road embankment, composed of loose sandy and sandy-loamy soils. The main reasons for the subsidence of the permafrost table are the transformation of the natural conditions serving the polygonal peatland ecosystem functioning. In roadside depressions, favorable conditions are created for waterlogging, the growth of tall shrubs, and snow accumulation as well [16]. The combination of these factors activates the thermokarst processes [45], resulting in a significant permafrost table retreat in the sites under road impact.

In natural regional ecosystems, the permafrost subsidence is less pronounced, in comparison with the zone of discontinuous permafrost in the European North. This is due to the greater stability of low-temperature, continuous permafrost in the North of Western Siberia. According to the classification scheme [46], the studied permafrost is classified as climate-conditioned, ecosystem-protected. In the case of ecosystem disturbances, permafrost in soil–geocryological complexes partially thaws, but repeated permafrost aggradation is also possible under favorable environmental conditions.

In this regard, it is necessary to pay attention to the protection and preservation of permafrost peatlands and to apply the concept of ecosystem services for the peatland use in the construction of infrastructure facilities. When designing and constructing the linear objects, it is required to arrange sufficient culverts to maintain the natural level regime of bog waters. Measures to prevent or reduce negative consequences on linear structures are the drawing up of grid lines of bog water runoff, determination of locations, and calculation of the size of culverts [44].

## 4. Materials and Methods

### 4.1. Study Site

The study area is located in the northeast of the Pur-Taz interfluve, 5 km west of the Gaz-Sale Settlement of the Tazovsky District of the Yamalo-Nenets Autonomous Okrug (Figure 8). This is an area of the subarctic southern tundra [47]. The climate of the region is cold-temperate with a moderate mean annual precipitation (517 mm) with long cold winters and relatively short, cool summers. The mean temperature of the coldest month of January is −25.7 °C, while July is +14.5 °C; 244 mm of precipitation falls in the summer time and September, while a stable snow cover forms in the first week of October and thaws only at the end of May–early June [48]. The averaged height of the snow cover is about 80 cm; however, frequent snowstorms (on average 54 snowstorms per a year) contribute to intensive snow transport, on average 378 m^3^/linear meter [49].

The study area is located in a subzone of predominantly continuous permafrost with a thickness from 400–450 to 250–300 m at the latitude of the Arctic Circle. Taliks are found only under the channels of large rivers and deep lakes. The upper permafrost has quite low temperatures (from −3 to −7 °C) and a significant distribution of syngenetic, ice-enriched (up to 40–60%) sediments, often in the form of thick ice–soil wedges [50]. Weak relief dissection and the permafrost occurrence result in the waterlogging. Ice wedges, flat peat plateaus, and polygonal peatlands are rather common for the region. Thereby, thermokarst shallow streams and depressions with small and large ponds arose from the thawing of ice wedges and were widely developed. Aeolian and abrasion processes intensively affect steep slopes and river and lake banks [51].

The study site is a polygonal peatland located in the III lacustrine-alluvial plain in the northeast of the Pur-Taz Interfluve and crossed by the Tazovsky—Gaz-Sale bulk road with a concrete pavement. This road was built in 2006 and, despite the presence of culverts, greatly changed the hydrological regime of the bog system under consideration, which led to intensive permafrost thaw, in particular in interpolygonal ice wedges and veins. Previous studies in 2016–2017 also revealed sharp changes in the topography of the peatland under study [13,52].

The studied peatland is not rare for the tundra ecoclimatic zone: The peat deposit thickness obviously exceeds 5 m and contains ice wedges with a vein height of 3–4 m. Similar thick polygonal peatlands were described earlier in lacustrine-bog depressions in the sites to the North, within the Yamal and Gydansky peninsulas [3,4,5].

### 4.2. Soil Sampling and Laboratory Analyses

In the field, geobotanical descriptions were carried out according to generally accepted methodology [53,54]. Soils were collected with a vertical resolution of 5–7 cm. The determination of soil types and the indexing of soil horizons were classified according to the International Correlation Base of Soil Resources [34]. In permafrost layers (PL), samples were collected using the method of manual drilling (a steel pipe hammered into the ground). In the active layer (AL), peat samples were taken in fixed-volume (503 cm^3^) cylinders inserted horizontally into the wall of a soil section. For peat and carbon accumulation calculations, dry bulk density (g cm^−3^) was measured contiguously for every 5–7 cm after freeze-drying volumetric subsamples and by dividing the dry mass (g) by the peat fresh volume (cm^3^). Carbon and nitrogen (C/N) content measurements were performed at 5–7-cm intervals, at the Institute of Biology (Syktyvkar, Russia), using an automatic CHN(S,O)-analyzer EA-1110 (Carlo Erba CE Instruments, Milan, Italy) and these results were applied to calculate average carbon values. To estimate the temporal variations in apparent carbon accumulation rates (ACAR, g C m^−2^ yr^−1^), the carbon mass of every 1-cm increment (g m^−3^) was multiplied by the corresponding vertical peat accumulation rate (m yr^−1^) based on the age depth models.

Plant macrofossil analysis was performed at the Institute of Biology Karelian Science Center, Russian Academy of Sciences. The samples were examined after deflocculating peat of known volume (5–20 cm^3^) with 5% KOH and sieving (150-μm mesh) to remove fine detritus; remains were identified under a stereo binocular (25–40× magnification) using reference literature [55]. Sphagnum species were identified by their leaf morphology [56] under the microscope (100–400× magnification). The abundance of selected species of plant macrofossils was calculated as a volume percentage of the total macrofossil assemblage. Visualization of gross stratigraphy and plant macrofossils were performed using the software «Korpi» (version 1.0) [57].

Spore-pollen analysis was done at the Institute of Geology of the Federal Research Center of the Komi Scientific Center of the Ural Branch of the Russian Academy of Sciences. Chemical processing of soil samples for palynological studies was carried out using standard techniques: alkaline Post and separation Grichuk, as well as Erdtman’s acetolysis technique [58]. Spores and pollen were identified using a digital biological microscope “Motic BA 300 (420× magnification). Spore-pollen diagrams were compiled using the “TILIA” software (version 1.7.16). Interpretation and calculation of the results of the spore-pollen associations were calculated in a group way. Spores and pollen in the spectra were combined into groups (pollen of trees and shrubs, grass pollen, spores), then the percentage of spore species and pollen grains was determined from 100% of the marked forms.

For radiocarbon (^14^C) analysis of bulk SOC, roots were removed under a microscope. The samples were treated with the standard method ABOx [59] at the Institute of Archaeology and Ethnography, Siberian Branch of Russian Academy of Science. The ^14^C concentration using accelerator mass spectrometry was measured at the Institute of Nuclear Physics of the Siberian Branch of the Russian Academy of Sciences. The results were calibrated by CALIBRE software (version 8.2, ^14^Chrono Centre, Queens University, Belfast) and IntCal20 calibration curve, and expressed in (median) calendar years before the present (cal BP; BP = AD 1950) (Appendix B: Figure A2) [60].

The permafrost table depth and the boundaries of the lithological contacts of the polygonal peatland were determined by the method of ground-penetrating radar sounding using a Zond-12E GPR (Radar Systems, Inc., Riga, Latvia) with a 300-MHz surface-shielded antenna connected to radar [16].

## 5. Conclusions

The paleoreconstruction carried out using spore-pollen and radiocarbon methods describes the development of a thick arctic peatland during the Holocene, including changes of vegetation cover, hydrological regime, and carbon accumulation. It shows that the peat accumulation began about 9814 cal. yr. BP, in the Late Preboreal (PB-2), at a rate of 1 to 1.5 mm/yr. During the Boreal, between 9000–8000 cal. yr. BP, less intensive peat accumulation continued (0.7 mm/yr). In the Atlantic, peat accumulation at first slowed down considerably: since 7749 cal. yr. BP, down to 0.26 mm/yr.; since 6801 to 5657 BP, down to 0.13 mm/yr.; and then it practically stopped. Compared to the tundra of the European Northeast, where, during the Holocene climate warming, the soil–geocryological complexes of peatlands repeatedly thawed, the soils of the studied peatland remained stable. Even taking into account the ongoing and projected climate changes, when both temperature and precipitation could increase all over region, arctic peatlands in continuous permafrost will remain protected from degradation. In case the hydrological regime is disturbed under anthropogenic changes, a rapid (over several years) degradation of permafrost and soil–geocryological complexes of polygonal peatlands will occur. Such changes might be irreversible and significantly transform the carbon balance of the region.

## Figures and Tables

**Figure 1 plants-10-02813-f001:**
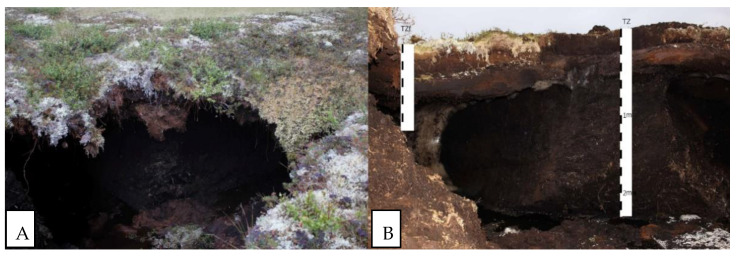
Vegetation cover (**A**) and TZ and TZf soils (**B**) developed in the peatland: Hemic Muusic Histosols (TZf) and Ombric Sapric Cryic Histosols (Hyperorganic) (TZ).

**Figure 2 plants-10-02813-f002:**
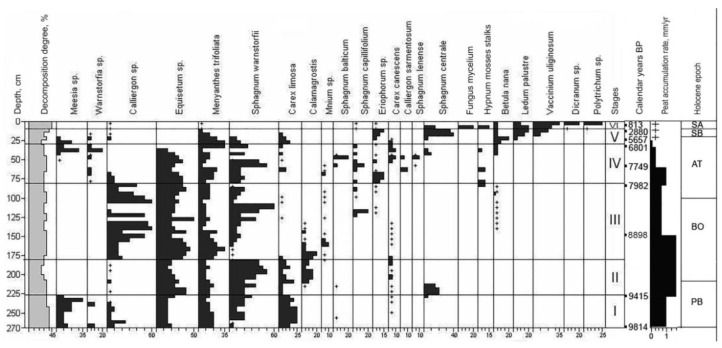
Plant macrofossil composition of TZ. +—0.5–2% of the sum of all species encountered. In the column “calendar age, years”, the dates 813, 2880, 8898, and 9415 are cited from the PhD thesis of Ya.V. Tikhonravova (laboratory numbers IMKES-14С1488, -14С1487, -14С1473, -14С1477, respectively) [35].

**Figure 3 plants-10-02813-f003:**
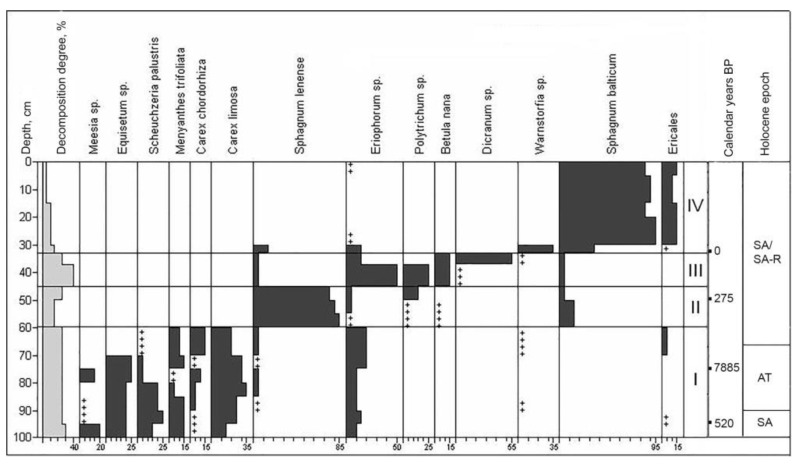
Plant macrofossil composition of TZf. +—0.5–2% of the sum of all species encountered.

**Figure 4 plants-10-02813-f004:**
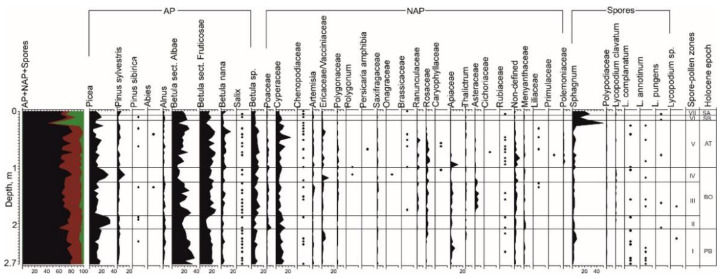
Spore-pollen diagram of TZ. •—0.5–2% of the sum of all species encountered.

**Figure 5 plants-10-02813-f005:**
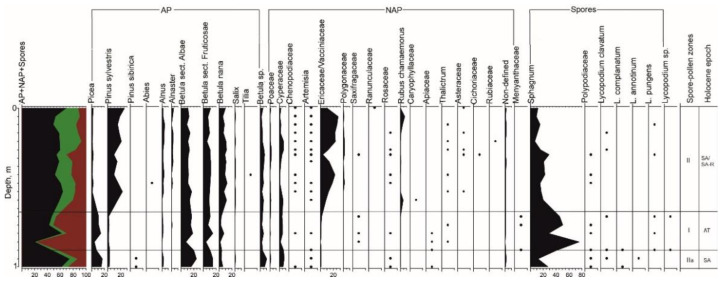
Spore-pollen diagram of TZf. •—0.5–2% of the sum of all species encountered.

**Figure 6 plants-10-02813-f006:**
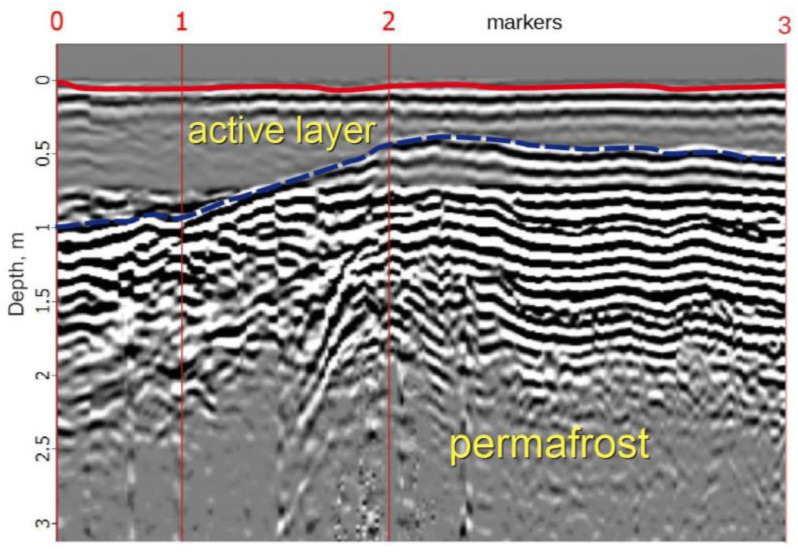
GPR profile 1 (roadside—undisturbed polygonal peatland) according to the data obtained with a 300-MHz antenna.

**Figure 7 plants-10-02813-f007:**
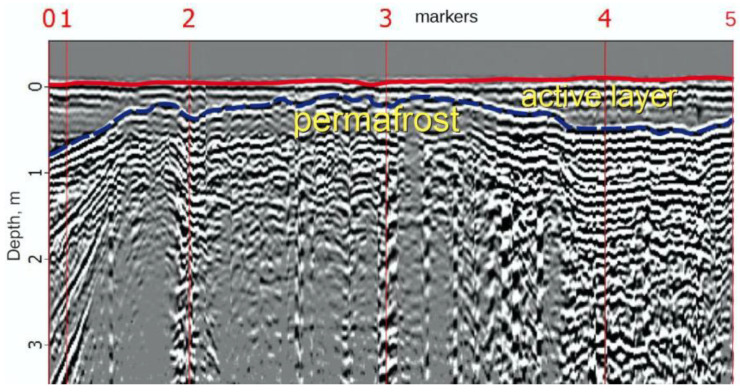
GPR profile 2 (thawing crack—polygonal peatland—crack) according to the data obtained with a 300-MHz antenna.

**Figure 8 plants-10-02813-f008:**
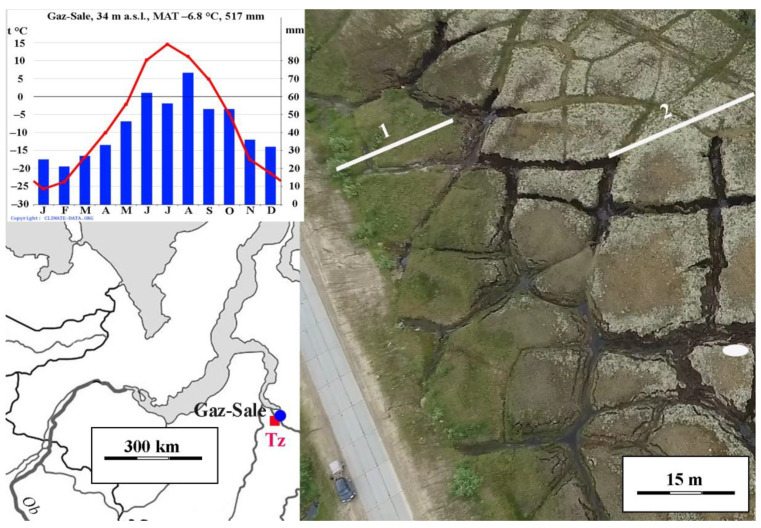
Location of the study area. Monthly temperatures and precipitation amounts based on the climatic database https://ru.climate-data.org/ (accessed on 18 October 2021) [48]. White lines indicate GPR profiles 1 and 2, and the white oval is the study site. Area survey was made from Phantom IV drone by E. Istigichev.

## Data Availability

The datasets used during the current study are available from the corresponding author on reasonable request.

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
