# Peer review of "Vulnerability of the Ancient Peat Plateaus in Western Siberia"

_plants, 2021, doi:10.3390/plants10122813_

Round 1

Reviewer 1 Report

A brief summary

A review of the manuscript entitled: „Vulnerability of the Ancient Permafrost Peatlands in Western Siberia”is very interesting and promising. In general, the manuscript is well written but I think that the authors need to make some insightful improvements.

Major concerns

  1. Calendar years and radiocarbon (14C) years were mixed up in the work (see Figures 2-5). As a result, the Holocene epoch has two different dates. How the peat increment was calculated? One C14 year may correspond to many calendar years. The older the holocene part, the bigger the differences. The paper needs considerable revision and a change in emphasis.
  2. The paper needs a lot of corrections in the results and the conclusions it follows mixed calendar years with radiocarbon years.

Based on this general evaluation and the specific comments, reported below, I recommend a major revisions of the manuscript and re-write and re-organization before it will be acceptable for publication. I have few specific comments, which might improve the manuscript.

New aspects should be emphasized.

Specific comment

Figure 4 - the fonts are not readable, mistakes in names: Polygonum amphibium according to The Plant List (http://www.theplantlist.org/) = Persicaria amphibia.

Figures are not cited in the text - „Error! Reference source not found”

Typos in plant names:

L 169 Carex limosa

L 173 Eriophorum vaginatum

L 180 Menyanthes trifoliata

Author Response

Response to Reviewer 1 Comments

Dear reviewer,

Many thanks for your great job and very valuable comments and remarks improving our manuscript.

Major concerns

Point 1: 1. Calendar years and radiocarbon (14C) years were mixed up in the work (see Figures 2-5). As a result, the Holocene epoch has two different dates. How the peat increment was calculated? One C14 year may correspond to many calendar years. The older the holocene part, the bigger the differences. The paper needs considerable revision and a change in emphasis.

  1. The paper needs a lot of corrections in the results and the conclusions it follows mixed calendar years with radiocarbon years.

In Figures 2 and 3 “Plant macrofossil composition”, errors in drawing the boundaries of Holocene Epoch are corrected. All dates are calibrated in the Calib software. The figures show only the median age. This is indicated in lines 532-535.

We add to Methods: For peat and carbon accumulation calculations, dry bulk density (g cm−3) was measured contiguously for every 5-7 cm after freeze-drying volumetric subsamples and by dividing the dry mass(g) by the peat fresh volume (cm3). Carbon and nitrogen (C/N) content measurements were performed at 5-7 cm intervals, at the Institute of Biology (Syktyvkar), using an automatic CHN(S,O)-analyzer EA-1110 (Carlo Erba) and these results were applied to calculate average carbon values. To estimate the temporal variations in apparent carbon accumulation rates (ACAR, g C m−2 yr−1 ), the carbon mass of every 1 cm increment (g m−3) was multiplied by the corresponding vertical peat accumulation rate (m yr−1 ) based on the age depth models.

Specific comments

Point 1: Figure 4 - the fonts are not readable, mistakes in names: Polygonum amphibium according to The Plant List (http://www.theplantlist.org/) = Persicaria amphibia.

Response 1: Figure 4 and 5, all mistakes were corrected.

Point 2: Figures are not cited in the text - „Error! Reference source not found”.

Response 2: Corrected

Point 3: Typos in plant names:

L 169 Carex limosa

L 173 Eriophorum vaginatum

L 180 Menyanthes trifoliata

Response 3: Typos in plant names were corrected.

Reviewer 2 Report

The article is about peatlands rather than permafrost, which should be included in the title (change the priority). The work presents original and interesting research results related to this issue. Especially dating provides interesting information. However, there is no broader link between peatland evolution and permafrost interaction / evolution. This is where Matti Seppala's publications from Finland can be of help. I highly suggest using them. More comments can be found in the text

Author Response

Response to Reviewer 2 Comments

Dear reviewer,

Many thanks for your great job and very valuable comments and remarks improving our manuscript.

Point 1: Line 2. The article is about peatlands rather than permafrost. Therefore, I suggest changing the title so that permafrost comes in second place.

Nevertheless, even with the proportions inverted, more attention is needed to permafrost: its extent, depth and evolution over the last 10 k years.

Response 1: It is. Permafrost peatlands, or permafrost-affected peatlands (peat plateaus), are found in high-latitude regions and store globally-important amounts of soil organic carbon. To avoid misunderstanding, I suggest changing the name to “Vulnerability of the Ancient Peat Plateaus in Western Siberia”.

Point 2: Line 28. “… general planetary processes”. Please, indicate more precisely.

Response 2: We added: “… general planetary processes, such as biogeochemical and biogeophysical cycles, greenhouse gases, activity and species diversity of vegetation and soil biota”.

Point 3: Lines 75-82. Instruction for authors?

Response 3: We deleted this paragraph

Point 4: Lines 87. (Error! Reference source not found.).

Response 4: Corrected

Point 5: Line 109. here is not much about state of permafrost in paragraph below, please give some more details as well as general informations see e.g. Dobinski W. 2020 Permafrost active layer, Earth-Sci Rev.

However, there is no broader link between peatland evolution and permafrost interaction / evolution. This is where Matti Seppala's publications from Finland can be of help. I highly suggest using them.

Response 5: We add the paragraph after line 109:

Permafrost peatlands are typical bog geosystems in permafrost environments in Canada (Zoltai 1995), Scandinavia (Seppala, 2003), European Russia (Pastukhov, Kaverin, 2016) and Siberia (Vasil’chuk Yu., Vasil’chuk A., 2016; Fotiev, 2017). Permafrost initiation involves peatland surface upheaval that results in drying of the peat surface, which is often prone to abrasion and erosion processes (Seppala, 2011). Scandinavian palsa mires (Seppälä, 2009) и Eastern European peat plateaus (Pastukhov, Kaverin, 2016) occur at the marginal zone of permafrost distribution. Therefore they may react rapidly on small changes in climate conditions like warming and increasing precipitation. But even with this, the small active layer thickness is caused by seasonal variations in the heat conductivity of the surface peat protects soil-geocryological complexes of permafrost peatlands from thawing (Dobinski, 2020). In contrast to palsas, the Western Siberian Arctic peatlands occurred in much more severe environmental conditions. Even a significant (by 10–15 °Ð¡) climate warming during the Holocene Atlantic optimum occurred on a vast area within the limits of negative temperatures and did not lead to the permafrost degradation. The climatic conditions of this epoch did not allow thawing of Arctic permafrost peatlands, reinforced with ice wedges. Ice wedges did not thaw from the surface even in the southernmost part of the Yamal Peninsula (67 ° N) (Fotiev, 2017).

Point 6: Line 111. Please do not use emotional expressions. Change may be only: fast/slow, big/ small. “dramatically”

Response 6: corrected to “rapidly”

Point 7: Lines 138. (Error! Reference source not found.).

Response 7: Corrected

Point 8: Lines 146. (Error! Reference source not found.).

Response 8: Corrected

We also corrected reference source for lines 161, 171, 178, 186, 196, 279

Point 9: Line 316. (Figure ) Fig. 6?

Response 9: Corrected – Figure 6

Point 10: Line 318. in fig 6 it is ca. 1m thick active layer maximum

Response 10: We added: “…the permafrost table depth varies in the range of 0.4–0.5 m (markers 2-3), while approaching to the road embankment, the permafrost table retreats to a depth of 1 m and below (markers 01)”.

Point 11: Line 329. description on the figure difficult to read. Need improovement.

What means numbers on x axis?

Response 11: We corrected Figures 6 and 7 according to recommendations. We added the name of x axis – “markers”

Point 12: Lines 341. (Error! Reference source not found.).

Response 12: Corrected

Point 13: Lines 389. (Error! Reference source not found.).

Response 13: Corrected

Point 14: Lines 463. Materials and Methods. This chapter should be placed as a second one. I do not understand why it is here?

Response 14: According to instructions for authors https://www.mdpi.com/journal/plants/instructions, we placed “Materials and Methods” after “Discussion”

Point 15: Lines 467. (Figure ).

Response 15: Corrected to Figure 3

Round 2

Reviewer 1 Report

The Authors provide sufficient answers to the questions. The Authors have made the appropriate changes in the main text and macrofossil diagram (Figure 2-5). I accept this version of the manuscript.

Reviewer 2 Report

I think the paper may be published in present form